# Value Chains and Diet Quality: A Review of Impact Pathways and Intervention Strategies

**Bradley Ridoutt [1,2,]*, Jessica R. Bogard [3], Kanar Dizyee [3], Lilly Lim-Camacho [3] and Shalander Kumar [4]**

[1] Agriculture and Food, Commonwealth Scientific and Industrial Research Organisation (CSIRO), Private Bag 10, Clayton South, VIC 3169, Australia
[2] Department of Agricultural Economics, University of the Free State, Bloemfontein 9300, South Africa
[3] Agriculture and Food, CSIRO, 306 Carmody Rd, St Lucia, Queensland 4067, Australia
[4] Innovation Systems for the Drylands, International Crops Research Institute for the Semi-Arid Tropics, Patancheru 502324, Telangana, India
[*] Correspondence: brad.ridoutt@csiro.au; Tel.: +61-3-9545-2159

**Abstract:** Low and middle-income countries increasingly face a triple burden of malnutrition encompassing undernutrition, micronutrient deficiency, and excessive energy consumption contributing to overweight and obesity. Food systems are also becoming more complex, supported by investments in food processing and retailing. Traditional approaches addressing hunger, typically based on agricultural development, are deemed insufficient alone to address the problem and attention is now being directed to food value chains, although experience is currently limited. To assess the state of science and identify knowledge gaps, an integrative review of the broad topic of value chains and diet quality was undertaken, with particular focus on interventions and their related impact pathways. Interventions were classified according to their primary orientation: to increase the availability, accessibility, or desirability of nutritious food. We identified 24 separate interventions associated with 10 different impact pathways, demonstrating the numerous entry points and large potential for value chain interventions to influence diet quality. However, case study evidence regarding effectiveness remains scant. Most studies addressed individual nutrient-rich commodities that could address a nutritional deficiency in the community of interest. Rarely was overall diet quality assessed, and future studies could benefit from taking a wider perspective of dietary patterns and food substitutions. The value chain analytical approach was deemed valuable due to its consumer orientation that seeks to understand how food products are used and what motivates their choice. The systems perspective is also important as it considers the roles of actors involved in food production, distribution, marketing, and regulation. However, few studies directly engaged with the subject of the local food environment as the bridge connecting food production and food choice. The challenge is to combat the increasing prevalence of processed foods of low nutritional value through interventions that lead to nutritious food becoming more conveniently available, affordable, and desirable.

**Keywords:** dietary diversity; food environment; food landscape; food system; nutrition-sensitive agriculture; triple burden of malnutrition; United Nations Sustainable Development Goal 2

## 1. Introduction

Malnutrition exists in various forms—undernutrition, micronutrient deficiency, excessive energy intake contributing to overweight and obesity—and is widely prevalent. According to the global nutrition report, no country is free from malnutrition [1]. Overall, progress in addressing the global triple burden of malnutrition has been mixed. While the global burden of undernutrition (dietary energy

deficits) has been impressively halved since 1990 and the global burden of micronutrient deficiency (hidden hunger) has reduced by around 30% [2], these gains have been largely offset by rapid increases in burdens associated with overweight and obesity [3], now impacting more than two billion people. The situation is complex because the different forms of malnutrition are inter-related. Insufficient energy intake is invariably linked to micronutrient deficiencies. However, diets characterized by excessive energy intake from saturated fat, sugar, and refined cereals, and diets lacking in adequate intake of fruits and vegetables and other nutrient-rich foods, are also commonly micronutrient deficient. The situation is further complicated by the co-existence of different forms of malnutrition within communities, especially in low and middle-income countries (LMIC) [4]. Individuals may even suffer from multiple forms of malnutrition, such as overweight and micronutrient deficiencies [5], and even simultaneous overweight and stunting in children [6], though the global prevalence of this burden is unknown [7]. These multiple burdens also interact across the life cycle, whereby stunted children are at an increased risk of overweight, obesity, and related non-communicable diseases later in life [8].

Traditionally, the problems of malnutrition in LMICs have been viewed through the lens of undernutrition, with priority given to increasing food production through agricultural development [9]. The cultivation of high-yielding varieties of major staple crops—wheat, rice, and maize—along with the development of irrigation infrastructure and the use of synthetic fertilizers, agricultural chemicals, and increased mechanisation (the green revolution) has led unquestionably to increasing the availability and reducing the cost of food energy, contributing in important ways to alleviating hunger, especially in Asia. For example, Food and Agriculture Organization (FAO) statistics show that global rice production has increased by more than 250% since 1960 (faostat.fao.org). However, by displacing traditional crops and reducing the price of major cereals relative to other nutrient-rich foods, such as fruits, vegetables, and pulses, there is evidence that dietary diversity may have inadvertently been undermined in some cases [10–13]. As such, it is unsurprising that progress toward addressing hidden hunger has lagged progress in addressing chronic hunger [14].

In parallel, food systems in many LMICs have also been undergoing major changes, with profound implications for nutrition [15]. Economic development and urbanisation have reduced the proportion of the population directly involved in agricultural production and the importance of traditional markets where local farm produce is traded. With rising incomes and less household food production, more complex food value chains have emerged, supported by investments in the food processing and retailing sectors [16]. Less physically active work, reduced average dietary energy needs, and increased consumption of highly refined carbohydrates and foods low in fibre and containing added sugar and salt, have led to widespread increases in the incidence of overweight and obesity across LMICs, and even as far as rural areas of East Asia, South Asia, and Sub-Saharan Africa [17]. With the progressive globalisation of the world food system, through the agency of trade, investment, and cultural influence, a large number of LMICs now share many of the same malnutrition challenges as developed countries.

Notwithstanding the important gains in food production quantity that have been achieved through agricultural productivity improvements, such as for rice and wheat in South Asia and maize in Sub-Saharan Africa, it is now recognised that there should be a wider focus on diet quality in LMICs [11,18–20]. This will be critical to address the United Nations sustainable development goals (SDGs), particularly Goal 2, which aims to end all forms of hunger and malnutrition by 2030, making sure all people—especially children—have access to sufficient and nutritious food all year round [21]. The multidirectional relationships among nutrition and other SDGs have also been underscored [22]. In the simplest food systems, local agricultural production and farm diversity have an important bearing on household and individual diet quality [23–25]. However, more complex food systems, characterised by longer value chains, larger-scale food processing and retailing, diminish the criticality of the local farming landscape by connecting consumers with agricultural production systems regionally and even internationally. As such, value chains have emerged as an important intervention pathway to improve nutrition.

For all of the above-mentioned reasons, the value chain approach to nutrition is suggested as important, but experience so far is rather limited [26]. Food value chains are distinguished from supply chains. The former encompasses all actors and their actions in creating products with attributes valued by consumers, whereas supply chains tend to relate mainly to the post farm-gate aspects of distribution. As such, farm production is considered within the scope of food value chains, which can be short, where there are few intermediaries between farmer and consumer, or long, where there may be many. In the extreme, a kitchen garden can be considered the shortest of all possible food value chains with production being local and with no intermediaries. Value chains incorporate the consumer perspective, so value chain interventions can also include activities to inform, educate, or influence consumer behaviour. Traditionally, value chain analysis has been applied in the context of consumer products and services, and applications in the agri-food sector are many (e.g., [27–30]). The underlying philosophy, introduced by Porter [31], is that the value chain, rather than individual firm, is the source of sustainable competitive advantage. As such, value chain analysis starts with understanding what constitutes value to the consumer and sets about coordinating the activities of value chain participants to efficiently deliver value added offerings. The concept has also been extended and used in the context of reducing value chain environmental impacts and increasing resilience [32], climate change adaptation [33–35], and food safety [36,37].

Value chain analysis has also been seen as a mechanism for poverty alleviation by connecting smallholder producers with higher value markets [20]. Through rural economic development, the potential then exists to indirectly impact nutrition if smallholder households use additional income to purchase food items that enhance dietary diversity [12]. While greater economic opportunities may accrue in some cases [38,39], there exists ample case study evidence demonstrating that poverty alleviation has not always followed global value chain integration [40–46]. In addition, higher levels of disposable income do not always translate into healthier food choices, as evidenced by the poor dietary patterns seen in many middle- and high-income communities in situations where food costs are a relatively small proportion of income. Strategies to integrate smallholder farmers with global value chains have also been criticised for not sufficiently considering the exposure of smallholders to risks, the power imbalances that can impact profit margins, and the implications of increased expenditure of labour by women in agricultural production [47].

None of these streams of value chain research address nutrition directly, largely because nutrition is not couched as a value derived from the value chain. Consequently, a new direction in value chain research has begun to emerge which questions the potential to impact nutrition through shaping the food environment. Unlike traditional product- or commodity-oriented value chain analysis, this new agenda takes a whole of diet perspective and views value chains as the link connecting agricultural production, and consumer food offerings. In this context, we performed a literature review of research studies concerning value chains and diet quality with specific attention to impact pathways and intervention strategies. We focused on diet quality as an important intermediate outcome contributing to improved nutrition. Changes in diet quality are generally more readily observed than changes in nutritional or health status, with the latter involving more complex data collection methods and requiring larger sample sizes and longer timeframes to observe any changes [48]. The aim was to present the state of scientific evidence and to identify important knowledge gaps.

## 2. Materials and Methods

The subject of value chains and diet quality is broad and amenable to a variety of study designs and mixed research methods. As such, an integrative review [49] was deemed appropriate to summarize both the current conceptual thinking as well as case study and other evidence. An original search for research articles was undertaken on 6 September 2018 using web of science (www.webofknowledge.com). The search terms "value chain*" OR "value-chain*" were combined with search terms pertaining to nutrition, diet quality, and dietary diversity, as well as expressions relating to the food environment (Table 1). The latter was included based on the a priori reasoning that nutrition-oriented value chain

interventions are likely mediated through changes to the food environment, which relates to the food sources and products surrounding people in their daily lives [50,51] as well as the wider physical, social, economic, cultural, and informational context impacting food choices [52]. No limitation was used in regard to publication date. The intention was to broadly survey the landscape of research articles on the topic. However, it is possible that not every relevant article was located.

**Table 1.** Search terms used in the review of value chains and nutrition. The asterisk serves as a truncation operator. Words match if they begin with the word preceding the * operator.

| Topic | Search Terms |
| --- | --- |
| Value chain | "value chain*"; "value-chain*" |
| Nutrition | "nutrit*"; "diet quality"; "diet diversity"; "dietary diversity"; "diet*" |
| Food environment | "food landscape"; "food environment"; "food map" |

The initial search identified 57 research articles after the removal of duplicate references. These articles were further screened for relevance. Articles were excluded if they were abstracts not supported by a full research paper, were in a language other than English, and if they concerned animal nutrition rather than human nutrition. Articles were included only where they described studies with improved dietary quality or nutritional status as a primary objective. This is in contrast to value chain studies having a predominant economic development objective with improved nutrition only as a potential indirect outcome. Finally, 33 research articles were included in the review. These articles were assessed in detail with the objective of discerning different types of nutrition-oriented value chain interventions and their related impact pathways. Articles describing case studies or research in a situational context were further evaluated to identify any generalizable findings.

## 3. Current Status of Knowledge

Value chains and diet quality was found to be a relatively new research topic. Of the 33 journal articles located, none were published prior to 2013 and the highest frequency of publication was in 2017 (10 articles) and 2018 (10 articles to September 6; Figure 1). The field appeared to be strongly oriented toward LMICs. Of the 25 studies having a specific national or regional focus, only one study concerned a developed country. Hattersley [53] used a nutrition-oriented value chain approach to study changes in the processed deciduous fruit market in Australia. Around half (i.e., 16) of the studies concerned countries in Sub-Saharan Africa, including Burkina Faso, Cote d'Ivoire, Democratic Republic of Congo, Ethiopia, Ghana, Kenya, Malawi, Mozambique, Liberia, Senegal, South Africa, and Uganda [54–69]. The next most frequently studied region was South Asia (9 studies), including Afghanistan, Bangladesh, and India [13,56,57,70–75]. One study concerned Peru [76], and eight other studies were not location specific [9,12,37,77–81].

Around half of the studies (i.e., 15) were diagnostic, whereby a value chain analysis was conducted, and recommendations were made in regard to interventions that could potentially improve the nutritional orientation. Six studies described specific value chain interventions (or combinations of interventions) that had a nutritional goal. Twelve of the studies were conceptual or described frameworks for nutrition-oriented value chain assessment. Around half of the studies had a particular focus on improving the nutritional status of vulnerable population subgroups. Other studies were oriented toward improving nutrition across the community generally. A wide range of animal and plant-based foods were studied. Animal-based foods included seafood, poultry, and dairy products. Plant-based foods included vegetables, fruits, groundnut, small millets, and grain legumes. Most of the studies concerned minimally processed foods. One study related to an iron-fortified sweet biscuit. Value chain interventions identified across the 33 located journal articles were subsequently classified as those primarily oriented toward increasing the availability of nutritious food, those primarily focussed

on increasing the accessibility of nutritious food, and others oriented primarily toward increasing the desirability of nutritious food.

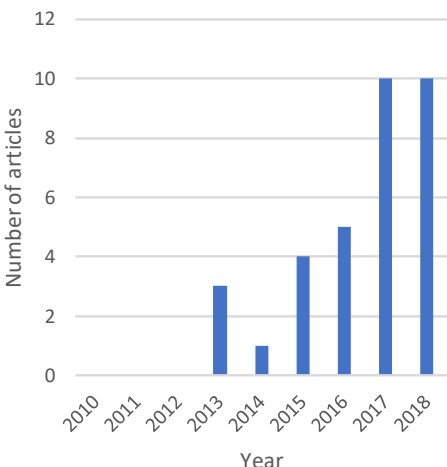

**Figure 1.** Histogram of located research papers according to year of publication.

### 3.1. Impact Pathways Related to Increasing the Availability of Nutritious Food

The physical availability of nutritious food in a local food environment is an essential prerequisite for nutritious food consumption. A dietary pattern can only be based on foods that are physically present. Several of the located literature described value chain interventions seeking to improve the production and supply of nutritious food, and thereby increase availability. Three main impact pathways were discerned (Table 2).

The first impact pathway related to the availability of nutritious whole foods, that might be increased by local or community production (short value chains) or by changes to distribution and retailing systems (longer value chains). Fan [57] described home gardening support programs in Bangladesh that led to increased local availability of vegetables, particularly leafy greens, and programs to support the cultivation of vacant land in the Democratic Republic of Congo that also increased local vegetable supply. In Uganda, the adoption of improved dairy cow breeds led to increased local milk supplies [62]. However, an increased availability of nutritious food might also be achieved by increasing the number of days or hours when such foods are able to be purchased. In some communities, vegetables are only available for purchase at informal markets with limited operating hours or from itinerant traders. This can limit the temporal availability. In Australia, the availability of processed deciduous fruits was increased though an expansion of retail channels to include petrol stations and convenience stores with extended trading hours [53]. The temporal availability of nutritious foods might also be increased through interventions that increase seasonal availability. This might be achieved through staggered planting systems, or systems of planting in different climatic regions [63], or by improved storage systems [57]. Processing of whole foods into ambient stable products that can be used year-round is another possibility, as demonstrated by the processing of sweet potato into flour [63].

**Table 2.** Summary of value chain interventions with potential to improve diet quality through increasing the physical availability of nutritious food in the local food environment.

| Impact Pathway | Intervention | Examples |
|---|---|---|
| Increase the availability of a key micronutrient though fortification | Bio-fortification of whole foods through new crop varieties | Bio-fortified orange-fleshed sweet potato to address vitamin A deficiency [63]. Bio-fortified iron beans [20] |
| | Mandatory industrial fortification of a food commodity | National Wheat Flour Fortification Programme in Pakistan [82] |
| | Voluntary industrial fortification of a processed food product | Iron-fortified glucose biscuits in India [61]. Iron-fortified yogurt in Senegal [64] |
| Increase the availability of nutritious whole food | Household and community production of fruits, vegetables and livestock products | Home gardening programs in Bangladesh [57]. Adoption of improved dairy cow breeds in Uganda [62] |
| | Changes in distribution and retailing that extend the number of hours and days when healthy whole foods can be purchased | Distribution systems that include permanent retail outlets, petrol stations, convenience stores and schools [53] |
| | Changes in the production and distribution systems that extend the seasonal availability of healthy whole foods | Staggered planting to extend harvest seasons [63]. Cold storage facilities used to reduce losses and increase the availability to consumers beyond the harvest-season [57] |
| | Packaging and processing to extend shelf life and facilitate wider distribution | Processing of sweet potato into flour that can be used year-round [63] |
| Improve the nutritional quality of processed food | Reformulation of processed foods to increase healthiness | Use of wholegrain ingredients with enhanced nutrient retention. Regulations to limit sodium content and trans-fats in processed food [61] |
| | Improve the affordability and therefore use of healthy foods as ingredients in processed food | Efficiencies to reduce the cost of millets to increase their use in the processed food industry [13] |

Another impact pathway focused on increasing the availability of key micronutrients in the local food environment is through fortification (Table 2). New varieties of crops can be developed, as demonstrated by the development of orange-fleshed sweet potato to address vitamin A deficiency [63], or the development of bio-fortified iron beans [20]. Fortification might also be applied during food processing. In some countries, mandatory vitamin and mineral fortification requirements exist for bread and other widely consumed processed foods. Ansari et al. [82] discuss the national wheat flour fortification programme in Pakistan, noting the potential for such programs to bypass vulnerable populations who may access food via informal value chains. There are also various examples of the voluntary fortification of processed food products, including breakfast cereals [61], yogurt [64], and sweet biscuits [72].

A third impact pathway involved improving the nutritional quality of processed food in the local food environment (Table 2). This might involve reformulation of processed food to make available products containing wholegrains, or lower levels of sugar and salt. Greenberg [61] notes that processed foods are often formulated in order to utilize low-cost ingredients. Interventions to reduce the cost of healthier ingredient options might therefore lead to greater inclusion in processed food. For example, efficiencies that reduce the cost of small millets have the potential to increase their use in the processed food industry [13].

### 3.2. Impact Pathways Related to Increasing the Accessibility of Nutritious Food

The physical availability of nutritious food in the local food environment is critical; however, this alone does not necessarily translate into high quality dietary patterns. Nutritious food must also be economically accessible. The inability to access nutritious food is an important cause of malnutrition in developing [83] and developed countries alike [84]. In Australia, lower-income households consume smaller quantities of fruits and vegetables and have less dietary diversity [85–88]. The price of fruits and vegetables relative to other energy-dense foods has also been identified as a barrier to consumption [85,86]. Similar evidence exists in the UK [89] and elsewhere [90,91]. In the located literature concerning value chains and diet quality, a variety of interventions were identified that were focussed on aspects of accessibility (Table 3).

Nutritious food can be made relatively more affordable through value chain interventions that increase competitiveness by reducing losses and waste (Table 3). For example, Gelli et al. [59] describe the establishment of a poultry vaccination system in Burkina Faso to reduce livestock deaths and improve production efficiencies. In Ethiopia, Parmar et al. [66] describe the potential for improved harvesting and handling practices for sweet potato that keep the skin intact and reduce losses. Wesana et al. [69] applied lean manufacturing principles in assessing opportunities to reduce losses in dairy value chains in Uganda. Value chains might also become more efficient through interventions which reduce transaction costs, such as public investment in road and telecommunication infrastructure [67]. Many nutritious foods are highly perishable, meaning that business transactions and the movement of produce must occur swiftly. In Malawi, the establishment of producer and trade associations was suggested as a means of addressing high business transaction costs in the value chains of traditional vegetables [55].

Other authors have stressed the importance of value chain interventions that reduce risks. Contract arrangements between farmers and retailers or processors have been suggested as a means of reducing uncertainty for farmers producing vegetables and animal-based foods, incentivising investment, and improving the coordination of supply with demand [77]. Value chains for nutritious food can also be made more efficient through the adoption of new technology. In particular, Adekunle et al. [13] identified the need for more efficient de-hulling equipment for small millets in India. Swamy and Dharani [74] drew attention to the important need for financial institutions to support longer-term investment in value chain infrastructure, in addition to short-term farm production loans. Governments can also alter the relative affordability of nutritious foods through taxes and subsidies on specific agricultural commodities and food ingredients [61].

**Table 3.** Summary of value chain interventions with potential to improve diet quality through increasing the accessibility of nutritious food in the local food environment.

| Impact Pathway | Intervention | Examples |
| --- | --- | --- |
| Increase the accessibility of nutritious food by making it relatively more affordable | Value chain interventions that increase competitiveness by reducing losses and waste | Poultry vaccination system to reduce morbidity [59]. Improved harvesting and distribution practices [66] |
| | Value chain interventions that increase competitiveness by reducing transaction costs | Investment in road and telecommunication infrastructure [67]. Promotion of producer and trade associations [55] |
| | Value chain interventions that increase competitiveness by reducing risk | Contract arrangements between farmers and retailers or processors [77]. Minimum support prices for agricultural commodities |
| | Value chain interventions that increase production efficiency through new technologies | Improved varieties of crops. Improvement in value chain financing [74] |
| | Taxes and subsidies on specific commodities and foods | Tax on sugar [61] |
| Economic empowerment of value chain actors responsible for household food purchasing | Value chain interventions that empower women | Improving women's access to credit [59] |
| Increase the accessibility of nutritious food by public distribution | School meal programs | The National School Nutrition Programme in South Africa [63]. School feeding programmes in Ghana and Nigeria linked to producers of orange-fleshed sweet potato [63] |
| | Food distribution to vulnerable individuals and households | Hot cooked meals provided to children attending *anganwadi* community centres in India [70] |

The accessibility of nutritious food can also be improved through economic empowerment of value chain actors responsible for household food purchasing (Table 3). In many cases, this involves the economic empowerment of women. Gelli et al. [59] describe a value chain project in Burkina Faso that is designed to improve diets through the development of poultry value chains that increase the income and purchasing capacity of women. Conversely, value chain interventions, when they involve enlargement of scale, intensification of production, and formalisation of contracts have been found to sometimes shift the control of revenues in favour of men, who may give a lower priority to spending on nutrition [60,77].

Public distribution is another intervention pathway to increase access to nutritious food by vulnerable population subgroups (Table 3). The development of value chains linking producers of orange-fleshed sweet potato to school meal programs in South Africa, Ghana, Nigeria, and Uganda have led to improvements in vitamin A intake among children [63]. In India, hot cooked meals are provided to children attending anganwadi community centres [70]. As part of this government-sponsored food distribution programme, the anganwadi centre workers are responsible for purchasing vegetables from local producers and markets.

### 3.3. Impact Pathways Related to Increasing Nutritious Food Choices

The physical availability and economic accessibility of nutritious food in a local food environment may not translate into high quality local diets. It is also necessary for individuals and persons responsible for household food purchasing and the preparation of meals to make nutritious food choices. The factors that influence individual decision-making around food are many and varied and have social, environmental, and psychological dimensions [92–95]. The research literature on value chains and diet quality included a wide range of interventions that aimed to improve the desirability of nutritious food (Table 4).

The desirability of nutritious food can be increased through value chain interventions that improve quality (Table 4). This is especially important for perishable fresh foods, such as fruits and vegetables, which are highly susceptible to quality losses. In the case of traditional vegetables in Malawi and Mozambique, weak linkages between supply chain actors led to quality losses [55]. Poor value chain integration can lead to delays in distribution of produce, poor coordination of supply and demand, as well as poor understanding and application of necessary handling and storage practices, all of which can result in poor quality offerings for consumers. Fruits and vegetables can also be susceptible to high levels of variability in size and quality, and if growers lack accurate market intelligence, consumer expectations can be poorly met. In Kenya, the size and colour of amaranth leaves were found to be key quality attributes influencing purchase decisions [65]. With meat and dairy products, which are susceptible to contamination, consumers may have concerns about food safety that can be addressed through value chain interventions focussed on hygiene and biosecurity practices [37].

**Table 4.** Summary of value chain interventions with potential to improve diet quality through increasing the desirability of nutritious food in the local food environment.

| Impact Pathway | Intervention | Examples |
| --- | --- | --- |
| Increase the desirability of nutritious food by quality improvement | Value chain interventions that develop or retain quality attributes valued by consumers | Leaf color and size influence the purchase of amaranth in Kenya [65]. Handling practices to retain the quality of traditional vegetables [55] |
| | Value chain interventions that provide consumers with confidence in food safety | Improved hygiene practices in livestock and fish value chains [37] |
| Education that improves knowledge about nutrition and the way nutritious food is valued | Education and communication on the benefits of nutritious food | Mass media, drama and direct contact used to increase demand for orange-flesh sweet potato in Mozambique |
| | Packaging and labelling to enable consumers to identify nutritious food | Nutritional labelling of take-home rations in India [70]. Private certification schemes [72] |
| Increase the desirability of nutritious food by improving the way its status is perceived | Advertising and sponsorship | Promotion of ready-to-eat fruit snacks in Australia [53]. The association of Bambara groundnut with fertility in Malawi [58] |
| Increase the desirability of nutritious food by improving convenience | Develop nutritious food products that are quick and easy to prepare | Development of ready-to-eat and ready-to-use products from millets in India [13] |
| | Development of convenient packaging sizes that suit a limited budget | Marketing of small and affordably priced sachets of dairy products [72] |

Education that improves knowledge about nutrition is another intervention pathway to improve nutritious food choices (Table 4). Consumers do not always understand the nutritional characteristics of individual foods and the dietary sources of critical micronutrients. For example, in India, awareness of the nutritional benefits of small millets was found to be rather low [13]. In Northern Senegal, major dietary sources of iron were poorly understood by mothers [64]. In Mozambique, value chain projects designed to increase consumption of orange-flesh sweet potato have included mass media, local drama, and direct contact with consumers. In Northern Senegal, a communication campaign focussing on anaemia prevention was included in a project that developed an iron-fortified yogurt [64]. In Bangladesh, efforts to develop dairy value chains were accompanied by a communication program designed to increase awareness of the benefits of milk consumption by women and children who were found to be consuming less than the national recommended levels [75]. Nutritional labelling on packaged foods can also assist consumers to identify nutritious foods. Take home rations, distributed to pregnant women in India as part of a supplemental nutrition program, were labelled with nutritional information to encourage consumption [70]. Processed foods that are micronutrient enriched or reformulated to improve nutrition may not be distinguishable from other processed food, unless suitable information is provided [80]. Value chains can also use advertising, sponsorships, and endorsements to influence food choices (Table 4). In Malawi, consumption of Bambara groundnut was promoted based on traditional beliefs associating it with fertility [58]. However, it has been noted that the most active promotion appears to be associated with energy-dense nutrient-poor highly-refined processed foods by the private sector [61]. This has led to the emergence of regulations and voluntary guidelines on responsible marketing of food to children [61].

The desirability of nutritious food can also be increased by improving its convenience (Table 4). The long cooking time required by grain legumes, requiring more energy and water for preparation compared to cereals, was identified as a barrier to their consumption and an opportunity for value chain innovation [56]. Adekunle et al. [13] described the development of ready-to-eat products, such as cakes, biscuits, and noodles, and ready-to-use products, such as soup mixes and flours, to make the consumption of millets more convenient. Packaging can also be developed to suit a wider range of eating occasions. In Australia, processed fruits were packaged in ways that made them suitable for inclusion in school lunches or for eating on the go [53]. In India, dairy products have been packaged in small sachets convenient for purchase by lower income groups [72].

### 3.4. Case Study Evidence

The case study evidence concerning value chains and nutrition is currently rather scant, with only six of the 33 research articles describing specific interventions and only five of these presenting the results of an evaluation. One article described a project at study design stage only [59]. The limited available evidence suggests that value chain interventions can indeed lead to diet quality improvements in target populations, although not in every case. In Uganda, the introduction of improved dairy cow breeds resulted in higher milk yields per cow, higher milk sales, and greater market integration [62]. These households had higher expenditure on food and there was evidence of lower childhood stunting. In Northern Senegal, a value chain model involving the distribution of iron-fortified yogurt to households supplying milk to the local dairy factory led to improved haemoglobin concentrations in vulnerable preschool children [64]. In north-eastern Afghanistan, efforts to build capacity among women in the dairy value chain showed some promising early results [73]. Participating households were significantly more likely to consume dairy products. However, for the value chain to be commercially sustainable and therefore scalable, further development was needed to compete with imported dairy products. In the central highlands of Peru, it was the smallholder households that already had more resources and enjoyed better dietary diversity that were best able to participate in profitable niche value chains for native potato varieties [76]. This finding highlights the need for special consideration of vulnerable population subgroups in the design of nutrition-oriented value chain interventions. In Liberia, value chain initiatives led to increased household economic

welfare [68]. Access to food also improved and childhood welfare indicators appeared to be trending in a positive direction. However, the authors note the potential for the nutritional benefits of value chain interventions to be undermined by the dynamics of individual households and the way they manage their finances.

## 4. Discussion and Conclusions

Traditionally, value chain analysis in the agri-food sector has adopted a product or commodity-oriented focus without specific regard for nutrition. The located literature on value chains and diet quality had a similar focus, predominantly on agricultural commodities (e.g., small millets, groundnut, poultry, indigenous vegetables) and occasionally food products (e.g., iron-fortified yogurt). The main point of difference was the explicit focus on a type of food with high nutritional value that could address a nutritional deficiency in the community of interest. As such, the most common indicators of success in the value chains and diet quality literature were observed to be much the same as those used in traditional value chain analysis, such as efficiency in production, success in delivering products with attributes identified as important to consumers, and overall market penetration. Did households consume more milk? Did they consume more vegetables? To some extent, nutrition-oriented or nutrition-sensitive value chain research is not clearly distinguishable from traditional value chain research applied to an individual nutrient-dense food. In relatively few cases was overall diet quality considered (e.g., [68,76]). While the increased consumption of a specific nutrient-dense food may well contribute to improved diet quality in a community, this approach seems too narrowly aligned with the traditional value chain model, concerned only about consumption of the product under analysis. There would appear to be scope for value chain research that is intended to impact diet quality to start with a more complete understanding of the dietary context, including how individual foods contribute to dietary patterns and how different food groups might contribute to improved nutritional outcomes. For example, if there was a goal to increase consumption of fruits, it would be helpful to understand, in a local food culture, the likely eating occasions for fruits. Also, if more fruits were consumed, what other foods might they displace and what might be the nutritional implications. Recent framework documents note the need for such a transition (e.g., [26]), but this has yet to be widely translated into practice.

That said, it is evident that the value chain analytical approach has much to offer as an integrating framework linking agricultural production and food consumption. Clearly, increased supply of nutritious food is not sufficient to improve diet quality, as consumers frequently make food choices that are not motivated by nutrition. Likewise, heathy food choices are less likely when the availability of nutritious food in the local food environment is limited or relatively less affordable. Poor nutrition is a complex problem, dependent on many factors, and beneficial interventions are not easily identified. When applied to the problem of poor nutrition, the value chain analytical approach is useful because of its consumer orientation that seeks to understand how food products are used and what motivates purchasing decisions. The approach also has a systems perspective that considers the roles of the various actors involved in food production, distribution, marketing and regulation. As such, the approach is multi-disciplinary and also has a practical orientation. Most of the specific interventions found in the value chains and diet quality literature were not considered novel (Tables 2–4). However, it was their application in the context of a value chain framework that was relatively new. The breadth of different interventions found in the literature, addressing availability, accessibility, and desirability of nutritious food, also highlights the numerous potential entry points. Indeed, the value chains and diet quality literature points to the need for multiple concurrent interventions, such as interventions to improve food safety combined with behaviour change communication on nutrition and health [59], interventions to enhance the micronutrient density of crops combined with the development of new distribution networks [63], or the combination of new product development and marketing [13], as examples.

This review of value chains and diet quality identified 24 different interventions that were associated with 10 different impact pathways (Tables 2–4). While this is not an exhaustive list of all potential interventions and impact pathways, it does demonstrate the diversity of strategies already emerging from the value chain analytical approach. However, somewhat unexpectedly, we found that relatively few of the located literature directly engaged with the subject of the local food environment, even though all of the identified impact pathways involve moderation of the local food environment in some way. Those articles addressing the local food environment highlight its importance as a bridge connecting food production and food choice [9,63]. Particular attention is drawn to the role of corporate strategies that shape the food environment through increasing the accessibility of low-cost processed food products that are desirable, but typically low in nutritional value [61,77]. Also notable is the early work of Gereffi and Christian [96] on global value chains, linking the increasing incidence of childhood obesity to the distribution and advertisement of energy-dense, nutrient-poor processed foods. Hattersley [53] notes that food and nutrition policies have generally failed to address the public health nutrition implications of these changes in the local food environment. For value chain analyses seeking to identify ways to impact diet quality, a more overt consideration of the local food environment would seem desirable. Perhaps lessons could be learned from the emerging evidence base concerning obesogenic environments [97,98]. Obesogenic environments encourage lifestyles characterised by poor nutrition and low levels of physical activity. For example, in obesogenic environments, the density of fast food outlets is high and the built environment makes active modes of transportation difficult. The challenge for nutrition-oriented value chain analysis is to identify ways of impacting local food environments such that nutritious food is conveniently available, relatively affordable, and desirable. This is why the transition from nutrition-sensitive agriculture to nutrition-oriented value chains is so important, because value chain analysis is concerned not only with the agricultural phase of the food system, but also with the transformation of commodities into food products as well as their placement, pricing, and promotion in the food environment.

Value chain analysis would appear to have an important role in informing interventions aimed at improving diet quality. However, the field is rather new and there would appear to be considerable scope for the approach to evolve further from its origins so that it is more distinctly suited to addressing malnutrition challenges. Food systems are complex. When land, water, and other scarce resources are used for the production of one type of crop, this necessarily displaces the production of others. When farmers focus their activities on cash crops, this increases household income, but may displace production of crops for home consumption. When women are more engaged in paid employment, they may have less opportunity for traditional home gardening and meal preparation. When children consume nutrient-poor biscuits and other treats, this may displace intake of fruits and other nutrient-rich whole foods. A value chain analytical approach that focusses on a single product or commodity therefore has its limitations in understanding the potential outcome for overall nutritional quality. System dynamic modelling is another analytical approach that can be aligned with value chain analysis and used to evaluate feedback effects among different components and sub-components of a system and understand trade-offs and system-wide consequences. The approach has recently been applied in the context of food systems [36,99,100] and has the potential to be applied more specifically in the assessment of nutritional outcomes of food system interventions. Similarly, food systems depend upon access to scarce environmental resources and operate in a national and global policy context that includes environmental goals, such as greenhouse gas emissions reduction (SDG 13 climate action). Life cycle assessment [101] is another analytical tool that can be aligned with value chain analysis to enable the environmental aspects of nutrition to be incorporated [102], providing linkage to other SDGs such as SDG 12 on responsible production and consumption.

A common feature of food environments in many LMIC countries is the increasing prevalence of processed foods that are of low nutritional value but are conveniently available, economically affordable, and highly desirable [15]. Isolated value chain interventions relating to particular nutrient-dense foods are unlikely to have a major impact on diet quality in such an environment. It is therefore necessary that

more attention is given to the policy and regulatory system that economically inclines that agricultural production system toward low-cost staple crop production, the food manufacturing sector toward the production of highly refined carbohydrate-based foods low in fibre and containing added sugar and salt, and the food distribution and retailing system toward ambient stable packaged foods. Value chain analysis is more commonly applied at the scale of individual products where unique commercial opportunities can be identified and exploited. However, the approach is not limited in its potential application at much larger scales that examine policies and regulations that overarch the food system. As governments grapple with the economic and social costs of increasing overweight and obesity, this would appear to be a further important opportunity for value chain analysis.

**Author Contributions:** Conceptualization and development of framework, B.R., J.R.B., K.D., L.L.-C., and S.K.; critical review of literature, B.R. and J.R.B.; writing—original draft preparation, B.R.; writing—review and editing, J.R.B., K.D., L.L.-C., and S.K.

**Funding:** This research received no external funding.

**Acknowledgments:** This work was supported by the CSIRO Agriculture and Food, Value Chain Analytics Platform initiative. S.K. acknowledges funding from the CGIAR Research Program on Grain Legumes and Dryland Cereals (GLDC) supported by CGIAR Fund Donors (www.cgiar.org/funders/).

**Conflicts of Interest:** B.R. and L.L.-C. have undertaken food systems research for a variety of private sector organisations and Australian government agencies. We declare no other competing interests.

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
