# Peer review of "Value Chains and Diet Quality: A Review of Impact Pathways and Intervention Strategies"

_agriculture, doi:10.3390/agriculture9090185_

Round 1
Reviewer 1 Report
In Abstract, please mention the focus of the article is LMICs (low and middle income countries).
In page 2, lines 60-70, please add data to prove your point. Global trend data would be enough.
In page 2, lines 71-82, please add data to show changes in LMICs. Please define which countries are in LMIC category.
In Page 2, lines 87-88, please check definition of SDG2. Please add Source for SDG2 definition.
In Page 3, after line 124, please add discussion of inefficiencies along the food value chains, shown through high margins from one node of value chain to another node of the same value chains, i.e. margins charged by traders/marketers.
Identifying 3 main categories of interventions is very illuminating and helpful for the discussion in the paper.
Please add Reference Numbers to the Examples in Tables 2, 3, 4.
Organization of Table 3 in page 9 into Impact Pathways and Interventions does not match text in Section 3.2 exactly. Please check writing in Section 3.2.
In section 4, paragraph 1, authors write “Not surprisingly, the most common indicators of success in the value chains and diet quality literature were much the same as those used in traditional value chain analysis, such as efficiency in production, success in delivering products with attributes identified as important to consumers, and overall market penetration.” This conclusion cannot be drawn from the literature review presented in earlier parts of the study. How did you come to this conclusion?
In section 4, paragraph 2, authors write “There would appear to be scope for value chain research that is intended to impact diet quality to start with a more complete understanding of the dietary context, including how individual foods contribute to dietary patterns and how different food groups might contribute to improved nutritional outcomes”. This is an excellent point. Would you consider expanding this point?
Page 13, paragraph 1, please add discussion on inefficiencies along value chains to the discussion of interventions already written.
In Section 4. Discussion and Conclusions, please add “data requirements for research on nutrition sensitive value chains”.
On page 14, lines 444-445, SDG13 and life cycle assessment are not linked to the analysis in the study. Please consider linking to SDG1 and SDG12 instead.
Reviewer 2 Report
I suggest you to cite the work of G. Gereffi and Chritian who, as noted by Morgan et al. 2018, "made the first substantive contribution to nutrition-oriented research on long chains when they presented a GVC approach to the study of childhood obesity at the WHO Early-Stage Expert Meeting on Trade and Healthy Diets"
"This paper proposes a review of the emerging literature concerning value chains and diet quality with specific attention to impact pathways and intervention strategies.
This is an interesting topic that has not been directly addressed by value chain research mainly because nutrition is not couched as a value derived from the value chain.
Authors identified and discuss a list of 24 different interventions that were associated with 10 different impact pathways.
Authors explain that isolated value chain interventions relating to particular nutrient- dense foods are unlikely to have a major impact on diet quality and conclude that more attention need to be given to policies and regulations that shape the modern, complex food system.
The paper is well written and easy to follow.
The literature review is not very large, but quite comprehensively addresses the papers dealing with value chains and diet quality.
I suggest to further elaborate on the way global food value chains shape consumption and diets and the interactions between global and local food value chains. See for example the papers by Gereffi and Christian on this issue. I guess the reference to the global food value chain literature would help the authors to better explain the concepts presented in lines 409-4012 and 450-458."
